# Functional Analysis of the Spine with the Idiag SpinalMouse System among Sedentary Workers Affected by Non-Specific Low Back Pain

**DOI:** 10.3390/ijerph17249259

**Published:** 2020-12-11

**Authors:** Éva Anett Csuhai, Attila Csaba Nagy, Zsuzsanna Váradi, Ilona Veres-Balajti

**Affiliations:** 1Department of Physiotherapy, Faculty of Public Health, University of Debrecen, 26 Kassai st., 4028 Debrecen, Hungary; varadi.zsuzsanna@sph.unideb.hu (Z.V.); balajti.ilona@sph.unideb.hu (I.V.-B.); 2Faculty of Public Health, University of Debrecen, 26 Kassai st., 4028 Debrecen, Hungary; nagy.attila@sph.unideb.hu

**Keywords:** SpinalMouse, low back pain, skin surface measures, occupational health, health promotion

## Abstract

WHO describes “low back pain” (LBP) as the most common problem in overall occupational-related diseases. The aim of this study was to evaluate characteristics of spinal functionality among sedentary workers and determine usability of the SpinalMouse^®^ skin-surface measurement device in workplace settings in a risk population for LBP. The spinal examination was implemented at National Instruments Corporations’ Hungarian subsidiary, Debrecen in October, 2015, involving 95 white-collar employees as volunteers to assess spinal posture and functional movements. Data from the physical examination of 91 subjects (age: 34.22 ± 7.97 years) were analyzed. Results showed significant differences (*p* < 0.05) in posture and mobility of the spinal regions in sitting compared to standing position. Significant positive correlations were observed between values measured in standing and sitting positions in all observed regions and aspects of the spine (*p* < 0.05) except posture of lumbar extension (*p* = 0.07) and mobility of sacrum/hip in E-F (*p* = 0.818). Significant (*p* < 0.001) difference (5.70°) was found between the spinal inclination in sitting 6.47 ± 3.55° compared to standing 0.77 ± 2.53 position. Sitting position has a negative effect on the posture and mobility of the spine among white-collar employees. The SpinalMouse can be used effectively to determine spinal posture and mobility in cross-sectional studies and impact analysis of physical exercise interventions.

## 1. Introduction

In recent decades, the importance of low back pain (LBP) has been highlighted by public health professionals to reduce the incidence and prevalence of this condition. According to the Global Burden of Disease Study (GBDS), LBP remained one of the leading causes of 301 diseases and injuries both in 1990 and 2015. The age of the risk population is between 25–64 years. According to the latest GBDS, the observed years lived with disability (YLDs) related to LBP exceeded expected levels in high income countries, with a mean increase in the number of YLDs of 24.8% from 1990 to 2006 and 18.0% from 2006 to 2016 [1,2,3]. Based on the analysis of the World Health Organization (WHO), “back pain” is the most common problem in overall occupation-related diseases with a proportion of 37% [4]. According to a report from the European Agency for Safety and Health at Work (EU-OSHA) 2000, LBP life-prevalence ranged from 59% to 90%, with a point prevalence of 15% to 42%, depending on the target population of the study and the definition of back pain, its annual incidence was close to 5% [5]. Results of the National Health Interview Survey in 2011–2013 in the United States revealed that the first and most prevalent reported cause of work-disability were also back and neck problems with a ratio of 30.3% (95% CI = 29.1–31.5) among working-age (18–64 years) adults in both genders [6].

Low back pain is not a clearly specified disease; instead, it is a symptom caused by variable sources. Scientific literature makes a distinction differentiates “specific” and “non-specific LBP” according to diagnostic criteria. Triage of symptomatic subjects should be performed via clinical diagnostics, special tests and imaging, therefore patho-anatomical cause can be determined as a non-spinal source (e.g., arteriosclerosis of abdominal aorta, hepatomegaly, etc.); as a specific spinal disorder (e.g., cauda equina syndrome, spondyloarthropathy, osteoporotic vertebral fracture, spondylolisthesis, etc.); or radicular pain, radiculopathy and spinal canal stenosis. All those remaining cases which cannot be diagnosed as specific diseases are in the class of non-specific low back pain. The ratio of non-specific LBP in the total amount of back pain is close to 90% [7].

Nowadays, several studies are focusing on occupational health, health promotion at workplaces as well as reduction of the problem. The 2020–2022 Healthy Workplaces Campaign on the prevention of work-related musculoskeletal disorders (MSDs) of EU-OSHA aims to contribute to the better management of MSDs among European companies and organizations as an increasing tendency can be seen in prevalence worldwide [8]. Major risks include posture related risk, especially awkward, static posture along with weight-lifting, spinal twist and rotation as well as vibration; therefore, targeted health promotion programs are required to assess ergonomic risks and prevent further increases in the prevalence and incidence of work-related MSDs, especially LBP [9]. Since long-term static sitting position is among the most important risk factors of non-specific LBP [10], and LBP is the most prevalent occupational MSD [4], targeted public health programs are necessary to reach the most endangered populations at their workplaces.

Factors introduced above led us to conclude that a specifically designed, preventive workplace training program should be constructed to reduce the occurrence of occupational low back pain. The first step in the process of analyzing the impact of such an intervention implemented in a workplace environment is to collect reliable data of the target population. Since LBP is a major concern around the world which needs an urgent solution, it is important to have a deeper and more precise view of the problem and identify the spinal characteristics of the target population with the fastest and easiest-to-implement spinal examination method available. We believed that the SpinalMouse^®^ device was an effective tool for collecting relevant data.

Several studies have showed the reliability, validity and effectiveness of the SpinalMouse^®^ (Idiag, Volkerswill, Switzerland) examination tool, a new and validated, non-invasive, computer-assisted wireless telemetry device for the assessment of the curvatures, mobility and functionality of the spine and it could therefore be adopted in workplace interventions. Livanelioglu et al. studied the validity and reliability of “SpinalMouse” assessment in the frontal plane among fifty-one adolescent idiopathic scoliosis patients by comparison with radiologic Cobb-angle measurements. The inter- and intra-rater reliability of the Cobb and SpinalMouse measurements were excellent (intra-class correlation coefficient—ICC = 0.872–0.962) and a strong or very strong correlation was found between the two measurement methods (*p* < 0.0001) [11]. The reliability of the device was also assessed by Kellis et al. among eighty-one children (male, 10.62 ± 1.73 years) with no history of spinal disorder. Measurements were performed in upright standing, full range of flexion and extension of the spine by three raters on two different days. Their results showed that the reliability (ICC) of thoracic- and lumbar curvature measurements ranged from 0.81 to 0.93; therefore, their protocol demonstrates good to high reliability [12]. In another study, published in 2004 to determine the validity and reliability of the skin-surface measurement device, Mannion et al. conducted an examination among twenty healthy subjects. Their results suggest that the SpinalMouse is a reliable tool for the evaluation of standing curvatures and ranges of motion of the spine; however, they do not recommend its use for the determination of intersegmental angles. However, the SpinalMouse was highly recommended for use in both research and clinical examinations, in workplace settings and in the areas of ergonomics and seating design for the measurement of the sagittal profile as well as overall and regional ROM of the spine [13]. A review of the literature by Barrett et al. referred to the findings of Kellis et al. (intra-rater ICC = 0.81–0.87; inter-rater ICC = 0.88–0.89) [12] and Mannion et al. (intra-rater ICC = 0.73–0.88; inter-rater ICC = 0.83–0.87) [13] as high quality researches and concluded that SpinalMouse has a very high intra- and inter-rater reliability [14]. Post and Leferink determined the inter-rater reliability of the device as they examined the posture and mobility of the spine in sagittal plane in 111 subjects in their study. Their results present that the SpinalMouse is a highly reliable device for the measurement of inclination in flexion and extension, also in total inclination. They divided their study group into two subgroups, 42 healthy and 69 “post-fracture of spine” volunteers. As the second group cannot be regarded as a group of healthy subjects, they do not recommend using their results as reference values of spinal range of motion (ROM) [15]. Topalidou et al. conducted their study among fifty patients with low back pain complaints to evaluate the test-retest reliability of the measurements performed by the tool both in sagittal and frontal plane. According to the statistical analysis of sagittal measurements, seventeen out of twenty-four parameters showed a high reliability with the highest correlation coefficient value (ICC = 0.90–0.99) and five parameters showed good reliability (ICC = 0.80–0.89), indicating that SpinalMouse is a reliable to monitor spinal posture and mobility in LBP patients [16]. Similar to the aforementioned study, Guermazi et al. also involved subjects affected by low back pain to assess validity of the results with the analysis of correlation between data collected by radiography and skin-surface measurements. The validity was rated as good to excellent for the overall mobility of the lumbosacral spine (CCS = 0.86) and for overall mobility of the lumbar spine (CCS = 0.7) [17].

Several authors have reported that the SpinalMouse can be used to assess impact of interventions performed among large numbers of subjects. The same device was used for the analysis of the effects of four different therapeutic methods on posture and mobility of the spine by Celenay et al., where lumbar and thoracic posture and mobility were evaluated in the sagittal plane among ninety-six university students. Based on their measurements statistically significant differences were observed both in sitting and standing positions compared to baseline data (*p* < 0.05) [18]. A single-blind randomized controlled case-control trial was performed by Feng et al. in 2015, Beijing among 164 middle school students with a thoracic kyphosis angle over 40°. Subjects were randomly assigned to intervention and control groups in which posture and mobility of the spine were measured by the SpinalMouse to determine the effects of a targeted training program. Pre- and post-intervention examination was performed in the sagittal plane in upright, flexion and extension position of the trunk. Researchers were able to detect significant differences in inclination of the spine in the intervention group (*p* = 0.002) [19].

Previous studies have shown that skin-surface examination of spinal posture and mobility can be fast, effective and reliable in determining the spinal characteristics of sedentary employees in workplace settings. Furthermore, it can also be used for the evaluation of the effects of targeted physical training in workplace health promotion programs.

The aim of this study was two-fold. On the one hand, it was to evaluate the characteristics of spinal functionality, and determine changes in spinal mobility and posture in sitting compared to standing position—especially among sedentary workers with a possible risk of developing LBP due to long-term static sedentary work. On the other hand, as a pilot-study, it was intended to determine the applicability and feasibility of the SpinalMouse non-invasive examination method in workplace settings in a risk population for LBP as a possible examination device in workplace health promotion programs in the future.

## 2. Materials and Methods

Our work was an observational, cross-sectional study. Data collection was performed between the 1st and 22th of October, 2015 at the Hungarian subsidiary of National Instruments Corporations’ (Austin, TX, USA) in Debrecen at NI Hungary Ltd. From a total of 1182 employees those workers who had experienced lumbar complaints in the previous year had been invited to participate in spinal examinations via radiation-free skin-surface measurement during working hours. Motivation letters and an online screening questionnaire had been sent out via the company’s electronic newsletters in September which included information about dates and venues of examinations, explanation of inclusion-exclusion criteria and the process of registration for the schedule, moreover questions to identify and select subjects who met inclusion criteria. Inclusion criteria were: white-collar (sedentary) worker of the company; acute presence of non-specific LBP or previous experience of it during the last twelve months; having no other known, diagnosed spinal, internal organ or other musculoskeletal disorders; age between 20 and 60 years. Exclusion criteria were: blue-collar worker; subject had no previous low back pain complaint; low back pain was diagnosed using imaging by a doctor as a specific spinal disorder (e.g., spinal canal stenosis, disc protrusion and herniation, degeneration of intervertebral joints, etc.); having other diagnosed internal organ or musculoskeletal disorder; age below 20 and above 60 years. On the first three days of the study—at different times of the day—volunteers of pre-selection were invited to an introductory presentation about the aims and process of the examination and were asked to fill in a self-administered questionnaire described below. Completion was not compulsory and was not a prerequisite for the physical examination. Participants were examined on two different days in the company’s medical room. A total of 147 employees fulfilled inclusion criteria, 122 attended the presentations, 58 subjects completed the questionnaire, 95 participated in the physical examination and data of 91 volunteers were analyzed (Figure 1). Three physiotherapy students assisted the examination, one welcomed and briefed the participant, one placed skin-marks on the spine with a body marker after the subject undressed and one assistant handled the computer software of SpinalMouse during the physical examination. Measurements were performed by an examiner who was an experienced user of the SpinalMouse device. All equipment, human-resource and assessment methods were the same on the days of physical examination.

### 2.1. Questionnaire

A selection survey was published online via the company’s electronic newsletter to collect employees meeting inclusion criteria who were available and interested in participation in the study. The Data Collection Questionnaire was self-administered and was completed voluntarily by employees participated in the introductory presentation and was used to assess perceived health; quality and quantity of physical activity; amount of daily sitting time; incidence of low back pain; triggering postures related to LBP. A total of twenty-three questions contained self-developed questions, along with questions based on the Hungarian version of the European Health Interview Survey [20].

### 2.2. SpinalMouse

Intersegmental mobility, overall and regional spinal ROM and posture were measured in the sagittal plane both in sitting and standing positions using a SpinalMouse device (Idiag, Volkerswill, Switzerland), a wireless electronic measurement device, a non-invasive skin-surface tool for computer-assisted imaging and radiation-free examination [15].

Before starting the examination, every subject was registered in SpinalMouse software with gender, age and randomly allocated study codes. After undressing the upper body, spinous processes were palpated, and C7 and S3 were marked with a body-marker. Subjects were asked to take three different positions both in standing and sitting positions: relaxed but erect (not corrected), maximal flexion and maximal extension of the spine in which measurements were performed. SpinalMouse was run paravertebrally along the spinous processes from C7 to S3 segments, making the system capable of recording the contour of skin above the vertebral bodies in the sagittal plane.

Standing Position:Neutral in standing: Subject was asked to maintain a relaxed position with the feet shoulder width apart, with straight knees and arms by the side, looking and facing straight horizontally towards the wall.Maximal flexion in standing: Subject was asked to flex the trunk with straight knees as far as possible with slow motion from segment to segment, aiming to touch the ground with fingertips.Maximal extension in standing: Subject was asked to cross arms in front of the chest and extend the trunk as far as possible, keeping the knees straight, without extension of the cervical spine.

Sitting Position:

Examination was performed on a height adjustable chair in order to maintain precise 90–90 degrees in ankle and knee joints.4.Neutral in sitting: Subject was asked to maintain a relaxed sitting position with the feet shoulder width apart, arms by the side, looking and facing straight horizontally towards the wall.5.Maximal flexion in sitting: Subject was asked to flex the trunk as far as possible with slow motion from segment to segment, hanging arms beside the legs.6.Maximal extension in sitting: Subject was asked to cross arms in front of the chest and extend the trunk as far as possible, without extension of the cervical spine.

No warm-up was performed before the examination and each test was done once [15].

Intersegmental angles and inclination of the spine were determined in each position in relation to a vertical axis perpendicular to the ground. According to raw data of the examination, relative positions of vertebrae were calculated by the ‘intelligent recursive algorithm’ of the software, and the total and segmental ROM and position of the spine and all 17 segments (Th1/Th2—L5/S1) were estimated. Raw data from measurements are shown in a table which contains six columns: values from upright (U), flexion (F) and extension (E) are presented in the first three columns, where kyphosis angles are positive, and lordosis angles are negative values. Calculated mobility values of the segments are presented in the last three columns: range of flexion from upright (U-F), range of extension from upright (U-E) and total range from extension to flexion (E-F). Every column contains gender and age specific reference intervals for each segment presented before and after raw data. The bottom of the table shows total inclination and segmental (thoracic-, lumbar- and sacral spine/hip) mobility of the spine.

### 2.3. Feasibility

The secondary outcome measure was feasibility of the examination, determined by adherence, response rate, participation rate and drop-out rate. Adherence was defined as the percentage of employees who responded to the open call; response rate was defined as the percentage of the attended employees who completed the questionnaire; participation rate was defined as the percentage of registered employees who attended the physical examination; drop-out rate was defined as the percentage of attended employees who attended but did not complete the questionnaire; did not attend in the physical examination; data cannot be analyzed due to bias.

### 2.4. Data Analysis

A priori sample size calculation was performed with a power level of 80% and an α level of 0.05 using means (0.52 and 3.02) and standard deviations (3.62 and 3.29) relating to the standing upright inclination of the spine from the study of Kellis et al. [12].

Relevant characteristics of participants were presented by cross-tabulation; Chi-squared test was used to check the association between categorical variables. The Shapiro–Wilk test was used to check normality. Due to the non-normal distribution of the variables medians and interquartile ranges (IQR) were determined, and Wilcoxon rank-sum test was applied to evaluate differences between medians. Spearman’s correlation was used as well. The data were processed using Microsoft Excel and Intercooled STATA version 13.0 software (Publisher: StataCorp. 2013. Stata Statistical Software: Release 13. College Station, TX, USA, StataCorp LP).

Statistical analysis of the data was performed on 91 cases; three subjects were excluded due to measurement bias and one due to age-group restriction. In order to determine the characteristics of the posture and spinal mobility of sedentary employees the study focused on spinal inclination (I), thoracic (T), lumbar (L), sacrum/hip (S/H) position and ROM. Intersegmental ROM was neglected due to a lack of available scientific evidence. Median values and IQR were determined for selected regions of the spine. The study also sought to find out how posture (e.g., inclination) and mobility were changed by positional changes from standing to sitting position; a Wilcoxon rank-sum test was performed to determine whether these changes were significant (Table 1). Distribution of the studied population in the three categories determined by reference intervals for measured parameters was evaluated, and the percentages of participants determined in the classes of “below reference”, “in reference” and “above reference” range (Table 2). The correlation between individual results based on the measurements of the total sample size was analyzed to determine whether relatively the same posture and mobility could be measured in sitting compared to standing position strength of correlation was determined according to Spearman’s rho classification: 0.00–0.19—“very weak”; 0.20–0.39—“weak”; 0.40–0.59—“moderate”; 0.60–0.79—“strong”; 0.80–1.0—“very strong. (Table 3). However, some patients were excluded during data cleansing due to age restriction (<20 years-1 subject) and measurement bias (3 subjects), the final sample could be analyzed without any further missing values.

Feasibility was analyzed using descriptive statistics.

### 2.5. Ethical Approval

The study was approved by the Ethics Committee of University of Debrecen (5103-2018) and participants gave informed consent.

## 3. Results

According to our calculated sample size (*n* = 62), an open call for the study was introduced to the employees of the company. After voluntary application, 95 people attended the physical examination, four subjects were excluded due to measurement bias and age group restriction, leaving us with the data of 91 (38 men and 53 women, with an average age of 34.23 ± 7.83 years, the oldest was 57 years old, the youngest 21 years old) subjects for analysis.

### 3.1. Questionnaire

Self-administered questionnaires were completed by 58 employees of the study group (response rate: 63.7%). Average duty time was 40 h weekly. Data analysis revealed that 97% of the respondents perceived their own health status as satisfactory, good or very good and thought that they can do a great deal for their own health despite the fact that 98.3% usually spend most of their working hours in a sitting position, 51.7% sit 4–8 h and 39.7% sit more than 8 h a day. They were also asked about the regularity of LBP and nearly one-third of respondents marked back pain 1–2 times a year, 20.7% marked 1–2 times in three months, 20.7% marked 1–2 times a month, 13.8% marked 1–2 times a week and 12.1% experience LBP continuously. Respondents were also asked to identify postures or activities which usually trigger lower back pain. Results showed that 37.9% of respondentsexperience pain in “sitting”, 20.7% in “any position”, 19.0% during “exercising”, 15.5% in “standing” and 5.2% in “lying” positions. Answers related to strengthening and endurance training distributed over a week showed that 29.3% of employees never perform any targeted physical activity, 24.1% do so only once a week, 13.8% twice a week and 19.0% three days a week. One question focused on the reasons why subjects do not perform any physical activity. In response to that question, 1.7% of respondents answered “I do not need any”, 3.4% marked “I do not have the required physical ability”, 3.4% marked “There is no available sport facility nearby” and 39.7% answered “I do not have enough time”.

### 3.2. Comparison of Posture and Mobility in Sitting and Standing Position

Results showed that different measuring positions cause significant changes in the posture and mobility of the spinal regions in almost all measurement aspects except for posture in lumbar flexion (*p* = 0.258) and sacrum and hip extension (*p* = 0.413). The analysis related to the inclination of the spine in sitting (median 7° (IQR 4–9°)) compared to standing (median 1° (IQR −1–2°)) position reveals a significant (*p* < 0.001) anterior inclination of the whole spine. Change in the angle of lumbar curvature in sitting (median −10° (IQR −19–−2°)) compared to standing (median −27° (IQR −33–−20°)) position was also significant (*p* < 0.001), therefore the angle of lumbar lordosis was decreased in sitting position. Although angles measured in flexion of the whole spine in sitting (median 85° (IQR 71–90°)) were lower compared (*p* < 0.001) with the values measured in standing (median 101° (IQR 92–115°)) position, results related to the lumbar region showed a similarity (*p* < 0.258) both in standing (median 25° (IQR 20–30°)) and sitting (median 23° (IQR 18–30°)) positions. A similar pattern can be observed in the measurement performed in extension. Inclination of the whole spine showed a significant difference (*p* < 0.001) in sitting (median −19° (IQR −24–−14°)) compared to standing (median −24° (IQR −29–−19°)) position, and the lumbar inclination also presented a significant (*p* < 0.05) difference in sitting (median −32° (IQR −40–−27°)) compared to standing (median −38° (IQR −47–−29°)) position.

Results of the analysis revealed that mobility of the whole spine was also different in sitting compared to standing position, where it was larger from upright to flexion range of motion (ROM) in standing than in sitting position, and the same can be said in case of lumbar flexion when sitting (median 35° (IQR 26–43°)) and standing (median 55° (IQR 50–61°)) angles were compared. Upright to extension ROM showed significant (*p* < 0.001) changes between sitting (median −24° (IQR −34–−17°)) and standing (median −13° (IQR −17–−7°)) angles of the lumbar region like in the ROM (*p* < 0.05) of the whole spine (sitting = median −26° (IQR −30–−21°)) (standing = median −24° (IQR −28–−18°)), but ROM in this movement direction was larger in sitting. Despite the smaller standing ROM in extension, complete sagittal range of motion was larger in standing position and the difference between the angles was significant (*p* < 0.001) in case of the whole spine and the lumbar region as well (Table 1).

Deviation of total inclination in upright standing and sitting positions was compared with the standard neutral position (0°) to determine the possible negative effects of the sitting position. Mean deviation from neutral in standing position was 0.77 ± 2.53° and in sitting position, 6.47 ± 3.55°, respectively. The difference (5.70°) between the two means was significant (*p* < 0.001).

### 3.3. Comparison of the Results to Reference Intervals

Distribution of the studied population determined in percentages of participants was found in the reference range and below or above it. An extremely low (14.29%) proportion of participants were in the required normal range in standing upright inclination and a relatively low ratio (30.77%) in sitting extension inclination.

Focusing on the inclination of the entire spine in standing position, we found that 87.71% of subjects in upright; 24.18% in flexion and 54.95% in extension position were below the reference range. Results measured in standing position in relation to the lumbar spine showed that 28.57% of subjects in upright, 56.04% in extension and 53.85% in extension-to-flexion were above the reference range and 25.27% of subjects in flexion and 32.97% in upright-to-flexion were below reference data. In case of the results acquired in sitting position, we found 41.76% of subjects below the reference range in upright, 19.78% in flexion and 65.93% in extension. Analysis related to the lumbar spine in sitting position showed that 58–68% of subjects were in the reference interval in all parameters, which means that the remaining ratio of subjects presented deviation from the required values (Table 2).

A significant positive correlation between values measured in standing and sitting positions was seen in all regions and aspects of the spine except posture of lumbar extension (rho = 0.1912; *p* = 0.07) and mobility of the sacrum/hip in E-F (rho = 0.0244; *p* = 0.818). A very strong correlation was found in the upright standing and sitting values of the thoracic spine (rho = 0.860; *p* < 0.001) and flexion values of the lumbar spine (rho = 0.923; *p* = <0.001). Results of the correlation are shown in Table 3.

### 3.4. Analysis of Feasibility

Adherence to the invitation to the study was 23.35% (276 employees) from a total of 1182 employees of the company. Since 49.9% of the employees were non-office workers, adherence in the target group (663 employees) was 41.62%. Subjects of the target group were selected via online survey therefore 46.7% were excluded. The remaining 147 employees were invited to an introductory presentation where participation rate was 83% (122 employees) and response rate for the questionnaire was 47.54°% (58 employees). The total length of the examination was determined by the company therefore all the possible appointments (for 104 subjects) were booked in the timetable by the included subjects, but the participation rate was 91.34% since finally 95 employees were able to attend in the given time slots. Drop-out rate was 52.45% in the completion of the questionnaire, 8.65% in the physical examination and 4.39% during data cleansing due to measurement bias.

## 4. Discussion

This study focuses on the evaluation of characteristics of the spine in sedentary employees using a relatively new, wireless and radiation-free skin-surface device. Comparison of our results with the literature is difficult because so far, no studies have been published on the use of the SpinalMouse with an emphasis on spinal characteristics among sedentary employees affected by low back pain. Misir et al. observed changes in lumbar and spinopelvic parameters among LBP developers induced by prolonged standing (PD) (*n* = 20) and non-LBP developers (non-PD) (*n* = 18) using radiographs in various sitting and standing positions. Angle of lumbar lordosis was significantly smaller in the sitting upright and sitting flexion postures compared to upright standing angles. Their results showed 30% lower lumbar lordosis angle (LL) in PD group and 29.7% lower LL in non-PD group as they found flexed lumbar spine and posterior rotation of the pelvis in sitting position [21]. Comparison of lumbar lordosis angles between sitting and standing positions was performed by Endo et al. among 50 healthy subjects using radiography imaging and their results showed a similarity to the previous study in the decline of the angle of lordosis where a 50% lower LL was observed in sitting compared to standing position [22]. De Carvalho et al. also performed radiography imaging to determine changes in the curvatures of the spine in sitting position compared to standing among eight healthy subjects and they observed a significant decrease of the lumbar lordosis and posterior rotation of the pelvis [23]. However, we are not able to compete with the results acquired using radiography, but we can confirm previous research findings. Concurring with previous results, we also observed a significant anterior inclination of the whole spine and a significant decrease in the lumbar lordosis angle in upright sitting position. The end-position angle in flexion and extension of the whole spine and lumbar lordosis was significantly lower in sitting compared to standing. The comparison of upright sitting posture to neutral position also showed significant anterior inclination; therefore, the aforementioned findings support the view that sitting position has a negative effect on the position and distribution of spinal curvatures and, by changing biomechanics, it may increase the mechanical load of spinal structures.

An interesting result was found in the values relating to mobility from upright to extension since both the whole spine and the lumbar region presented larger angles in sitting position. Analysis of the position of the center of gravity (CG), the size of the supporting surface (SS) in sitting and standing position shows that it is logically understandable that the SS is larger and the CG is closer to the SS in sitting; hence, a larger range of extension can be performed with anterior tilt of the pelvis keeping the CG and body mass of the trunk above the SS with a lesser amount of required muscular effort.

Although we observed decreased mobility and lumbar lordosis angles with anterior inclination of the spine in sitting position, less than half of the participants presented upright inclination values below the reference range in sitting, but almost 90% showed upright inclination values below the reference range in standing. In relation to the reference interval given by the SpinalMouse device, we agree with the conclusion of Post et al. that further research is required to determine validated reference ranges classified by age, gender, height, weight and health condition. The relevance of the comparison to the reference data is questionable therefore these results can be disregarded [15].

We were also interested in determining the direction and strength of correlation of individual values acquired in sitting and standing position. Almost all variables showed statistically significant correlations suggesting that long-term habitual poor and awkward posture will change the standing posture of the spine and its regions; however, a strong or very strong correlation was found only in a few aspects of the thoracic and lesser of lumbar region. The same applies to mobility as the lesser the ROM in the sitting position, the lesser it will be in the standing position. According to our results, in accordance with findings by other authors, it can be concluded that upright sitting compared to standing position increases the anterior inclination of the whole spine, decreases the lumbar curvature, facilitates posterior tilt of the pelvis, increasing the mechanical load of the spine [23,24].

Our interest in the feasibility of such an examination led to a number of important results. Aiming to find the easiest and fastest method to examine a large number of subjects in a relatively short period of time, we chose SpinalMouse for the evaluation of posture and mobility. Time is a focal point in a workplace health promotion program since employees have to participate in interventions and examinations during working hours. Understandably, all activities, interventions, etc. have to be performed effectively in as short a time as possible. We found mass adherence to the physical examination among white-collar employees, after exclusion an extremely large ratio of subjects participated in our presentation. Unfortunately, the drop-out rate was relatively high in the completion of the questionnaire, since it was completed at the end of the presentation and a number of employees started to leave the room to hurry to the next meeting. In order to increase response rate, we suggest completing these questionnaires at the beginning of such a presentation or immediately before the physical examination. After the presentation at the company, an online 4-h schedule was published in which—with a 10-min gap—24 people were allowed to register, taking into account their own workplace responsibilities. Timetable slots were booked completely but, in the end, only 18 people attended. Based on the experience of the first examination day, in order to increase the number of participants, we set a shorter examination interval for the second day of the study, allowing 80 people to register with a 6-min difference during an 8-h workday, which resulted in the participation of 77 subjects. According to our experience, examination with the device in the sagittal plane both in sitting and standing positions took seven to nine minutes per capita on average. Reduction of required time for the examination can be attained by a precise timetable; preparation of the subject (undressing, marks on the skin on the spinous processes with the help of an assistant) should be performed outside the examination room. Since the device is not velocity-sensitive and the investigator will in time become more familiar with the tool, examinations can be performed faster. On the second day of the physical testing, we managed to reduce the time required for the SpinalMouse examination to less than 7 min. Participation rate in physical examination was extremely high (91.34%) thanks to the cooperation of the employer; the coordinated work of assistants; the strict adherence of subjects to the timetable. Those subjects who failed to appear at the venue of the examination subsequently indicated that a prolonged meeting was the reason and sadly, these accidental circumstances cannot be managed either by observers, the employer or the employee. The same statement is true in case of physical characteristics of the subjects, since it could be a source of measurement bias, which—in rare cases—is a limitation of the device.

Based on the findings in relation to the responses to our questionnaire, it can be concluded that a large number of employees work eight or more hours in a static sitting position which commonly causes LBP symptoms. In addition to the occupational risks, one-third of the employees are physically inactive, which was explained mainly with lack of time. These facts support the legitimacy of targeted workplace health-promotion interventions to reduce the health risk of sedentary lifestyle, inactivity and the incidence and prevalence of occupational LBP. An analysis was performed to compare results related to different measurement positions to evaluate the effect of the sitting position on posture and mobility of the spine. The findings confirm that a sitting posture has a negative biomechanical effect on the spine, leading to a reduction in the load absorption ability of the spine and to an increase in pressure on the intervertebral disc up to 40% in unsupported sitting, not to mention forward leaning and weight lifting with the increase by more than 100%, and the position of forward flexion and rotation with the increase by 400% [25,26].

Based on these findings, it is clear that inadequate ergonomics and poor posture in the sitting position present serious risks. The results of the study show that a relatively neutral upright standing lordosis of the lumbar spine and a biomechanically correct posture cannot be maintained in sitting position, hence a combination of the posterior tilt of the pelvis; sitting on os cocccygis instead of tuber ischiadicum; shortness of the hamstring muscles increases lumbar flexion; changes in the curvatures of the spine, the position of the head and neck; increase of disc pressure and eccentric load of the paravertebral musculature can be a possible causing factor of non-specific low back pain [24].

## 5. Conclusions

Based on our findings and those of previous studies on the posture and mobility of the spine, sitting position has a negative effect on the biomechanics and mechanical load of the spine among sedentary employees affected by LBP. Reliability and validity of the device have been studied by different researchers, and we also found that the implementation of the examination was fast, easy and efficient; hence, we recommend the application of skin-surface measurement in workplace health promotion programs. The SpinalMouse device can be used effectively to determine spinal posture and mobility in cross-sectional studies and impact analysis of physical exercise interventions in the future.

Based on the findings in the study, a specific, targeted, preventive physical and ergonomic training program is highly recommended in workplace settings to reduce the risk of the development of spinal disorders, low back pain and musculoskeletal diseases among sedentary employees.

### Limitations

During the measurements performed with SpinalMouse, we encountered a few weaknesses of the device as described by Leferink et al. [15]. The observer has to notice that marks on the skin presenting C7 and S3 spinous processes move with the skin in flexion and extension but do not follow the spinal column precisely; therefore, we suggest manual palpation of reference points during the examination to secure collection of valid data. Another possible measurement bias can be encountered in the case of sharp lumbar angles in upright and extension positions. A larger amount of soft tissue and extremely increased lordosis limits the possibility to follow the spinal column with the wheels of the SpinalMouse device; therefore, valid data cannot be obtained in these specific and relatively rare circumstances. Weaknesses of the self-administered questionnaire included the recollection bias on circumstances and events during the last twelve months and the response rate of participants since they were not as interested in completing a survey as in the physical examination itself.

## Figures and Tables

**Figure 1 ijerph-17-09259-f001:**
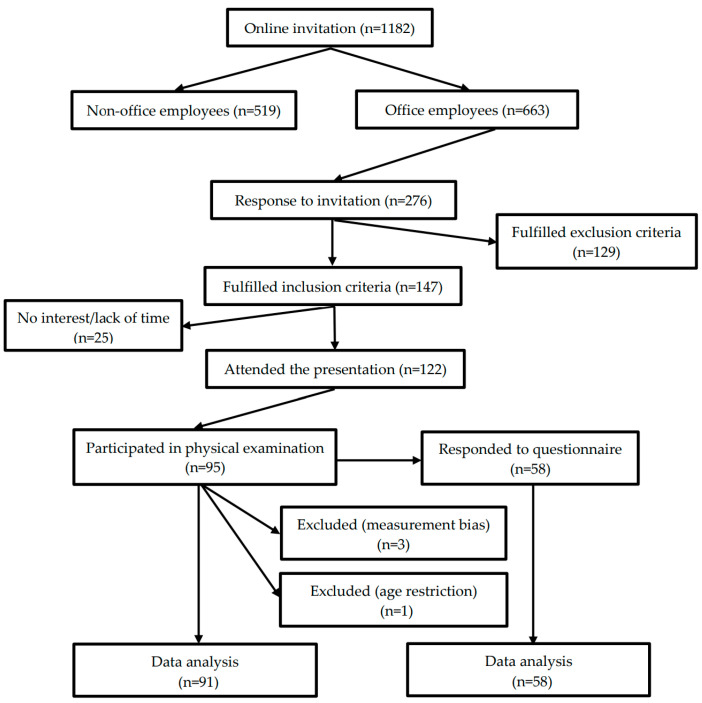
Flow diagram of selecting subjects for the study.

**Table 1 ijerph-17-09259-t001:** Median and interquartile range of total inclination, lumbar-, thoracic- and sacrum/hip inclinations in degrees measured in standing and sitting positions in upright (U), flexion (F), extension (E) and estimated range of motion from upright to flexion (U-F), upright to extension (U-E) and extension to flexion (E-F) (*n* = 91).

Position	Segment	Sitting			Standing			*p*-Value
		Q1	Median	Q3	Q1	Median	Q3	
U	Inclination	4	**7**	9	−1	**1**	2	<0.001 *
	Lumbar	−19	**−10**	−2	−33	**−27**	−20	<0.001 *
	Sac/Hip	1	**8**	14	9	**16**	21	<0.001 *
	Thoracic	27	**33**	43	35	**42**	50	<0.001 *
F	Inclination	71	**85**	90	92	**101**	115	<0.001 *
	Lumbar	18	**23**	30	20	**25**	30	0.258
	Sac/Hip	36	**45**	56	53	**65**	79	<0.001 *
	Thoracic	58	**64**	72	53	**60**	67	0.005 *
E	Inclination	−24	**−19**	−14	−29	**−24**	−19	<0.001 *
	Lumbar	−40	**−32**	−27	−47	**−38**	−29	0.033 *
	Sac/Hip	−1	**6**	10	2	**6**	12	0.413
	Thoracic	17	**27**	37	22	**31**	40	0.029 *
U-F	Inclination	65	**79**	85	91	**101**	113	<0.001 *
	Lumbar	26	**35**	43	50	**55**	61	<0.001 *
	Sac/Hip	31	**38**	46	41	**48**	64	<0.001 *
	Thoracic	23	**29**	36	10	**17**	24	<0.001 *
U-E	Inclination	−30	**−26**	−21	−28	**−24**	−18	0.049 *
	Lumbar	−34	**−24**	−17	−17	**−13**	−7	<0.001 *
	Sac/Hip	−8	**−1**	6	−15	**−10**	−6	<0.001 *
	Thoracic	−14	**−6**	−2	−19	**−11**	−7	<0.001 *
E-F	Inclination	87	**102**	113	113	**124**	141	<0.001 *
	Lumbar	50	**60**	67	60	**67**	74	<0.001 *
	Sac/Hip	31	**40**	52	48	**59**	73	<0.001 *
	Thoracic	29	**37**	44	20	**28**	36	<0.001 *

* significant result (*p* < 0.05); negative values are presenting lordotic angles, while positive values presenting kyphotic angle of the spine; bold indicates median values of study population.

**Table 2 ijerph-17-09259-t002:** Distribution of study population in percentages (%) among three categories (below reference range; in reference range; above reference range) in upright (U), flexion (F), extension (E), upright to flexion (U-F), upright to extension (U-E) and extension to flexion (E-F) characterized by total inclination (I), lumbar (L), thoracic (Th) and sacrum/hip (S/H) inclination measured in standing and sitting positions (*n* = 91).

Position	Categories	Standing				Sitting			
		I	L	S/H	Th	I	L	S/H	Th
U	below	85.71	5.49	10.99	18.68	41.76	13.19	7.69	24.18
	ref	**14.29**	**65.93**	**86.81**	**56.04**	**57.14**	**67.03**	**67.03**	**52.75**
	above	0	28.57	2.2	25.27	1.1	19.78	25.28	23.07
F	below	24.18	25.27	12.09	17.58	19.78	29.67	10.99	28.57
	ref	**61.54**	**71.43**	**62.64**	**50.55**	**72.53**	**63.74**	**69.23**	**53.85**
	above	14.29	3.3	25.27	31.87	7.69	6.59	19.78	17.58
E	below	54.95	0	37.36	41.76	65.93	12.09	15.38	23.08
	ref	**45.05**	**43.96**	**60.44**	**56.04**	**30.77**	**68.13**	**79.12**	**67.03**
	above	0	56.04	2.2	2.2	3.3	19.78	5.5	9.89
UF	below	16.48	32.97	13.19	12.09	19.78	32.97	18.68	21.98
	ref	**58.24**	**64.84**	**58.24**	**64.84**	**67.03**	**58.24**	**67.03**	**60.44**
	above	25.27	2.2	28.57	23.08	13.19	8.79	14.29	17.58
UE	below	5.49	25.27	8.79	2.2	1.1	10.99	5.5	10.99
	ref	**74.73**	**71.43**	**50.55**	**61.54**	**42.86**	**68.13**	**58.24**	**71.43**
	above	19.78	3.3	40.66	36.26	56.04	20.88	36.26	17.58
EF	below	26.37	1.1	40.66	41.76	24.17	9.89	29.67	20.88
	ref	**63.74**	**45.05**	**52.75**	**54.95**	**64.84**	**61.54**	**59.34**	**61.54**
	above	9.89	53.85	6.59	3.3	10.99	28.57	10.99	17.58

Bold indicates the ratio of the study population found in reference interval.

**Table 3 ijerph-17-09259-t003:** Correlation of individual values (degrees) measured in standing compared to sitting positions in upright (U), flexion (F), extension (E), upright to flexion (U-F), upright to extension (U-E) and extension to flexion (E-F) in relation to total inclination, lumbar-, thoracic- and sacrum/hip inclination.

Inclination		Lumbar		Sacrum/Hip		Thoracic	
r	*p*-Value	r	*p*-Value	r	*p*-Value	r	*p*-Value
Upright Standing—Upright Sitting							
0.238	**0.023**	0.370	**<0.001**	0.620	**<0.001**	0.860	**<0.001**
Flexion—Flexion Sitting							
0.5190	**<0.001**	0.9227	**<0.001**	0.5508	**<0.001**	0.7700	**<0.001**
Extension Standing—Extension Sitting							
0.4660	**<0.001**	0.1912	0.069	0.2446	**0.019**	0.7158	**<0.001**
U-F Standing—U-F Sitting							
0.4984	**<0.001**	0.5332	**<0.001**	0.2919	**0.005**	0.5845	**<0.001**
E-F Standing—E-F Sitting							
0.3773	**<0.001**	0.3202	**0.002**	0.0244	0.818	0.3727	**<0.001**
U-E Standing—U-E Sitting							
0.5596	**<0.001**	0.6506	**<0.001**	0.3065	**0.003**	0.4997	<0.001

r = correlation coefficient; *p*-value = level of significance (*p* < 0.05); r = Spearman’s rho; bold indicates significant result.

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
