# Peer review of "Functional Analysis of the Spine with the Idiag SpinalMouse System among Sedentary Workers Affected by Non-Specific Low Back Pain"

_ijerph, 2020, doi:10.3390/ijerph17249259_

Round 1

Reviewer 1 Report

The paper is well written, introduciton is exaustive and clear, really good explanation of statical analysis. Just two observation:

  1. please add the power analysis to justfy the number of subjects involved. I'm quite sure that the number of subjects involeved is enough but to be more accurate it could be better to insert a calculation (Altaman or other)
  2. is it possible to add some pictures of the subjects during the test?

Reviewer 2 Report

Dear Authors,

It was a pleasure to review manuscript entitled “Functional analysis of the spine with the Idiag SpinalMouse System among sedentary workers affected by non-specific low-back pain”.

Please find my comments below.

Background:

  1. Please shorten the introduction and include the scientific background and rationale for the reported study.

Materials and methods:

  1. How was the study design?
  2. When was the survey conducted? In. 2015?
  3. How were the eligibility criteria for participants?
  4. How was the study size calculation? Please add the final participants number in this section.
  5. Please add the flow diagram for study participants.
  6. Did the examination of 95 participants last 5 days?

Please organize this section using STROBE Statement checklist.

Results:

  1. “Prior to the study, a written informed consent was signed by each subject” Please do not repeat this information. Is should be in ethical approval section.
  2. Please do not repeat the statistical methods used in the results. They should be found in data analysis section.

Discussion:

  1. The discussion should not be a repetition of the results. Reference should be made to the results of already existing similar studies.

Conclusion:

  1. “Regional posture and mobility data obtained using this tool is objective and reliable and can thus be used as an outcome indicator of workplace physical and ergonomic training programs via pre- and post-intervention examination” In my opinion, this conclusion does not result from the conducted research.
  2. Please rethink the purpose of the study and reformulate the conclusions.

Reviewer 3 Report

Thank you for submitting your manuscript Functional analysis of the spine with the Idiag Spinal Mouse System among sedentary workers affected by non-specific low-back pain”. The manuscript is an interesting read. It details the negative association of sedentary life on spinal posture and mobility. A consequence of modern life. I would recommend strengthening the paper by ensuring all the aims are reported on, specifically, the secondary aim of feasibility, and making the introduction concise. Some are suggestions are detailed below.

Abstract

Results. The results describe ROM values, specifically “posture and mobility of the spinal regions in sitting compared to standing position” but then concludes “Sedentary life has a negative effect on the posture and mobility of the spine among white collar employees”. However, there is no information to draw this conclusion.

Introduction

The introduction includes 20 paragraphs with some paragraphs are only one sentence long, and hence are not a paragraph. At times like feels like the bullet points were written out in full sentences.

The content of the introduction includes multiple concepts throughout but does not provide a coherent story e.g. paragraph 1 and 10 is about low back pain, then sedentary lifestyle, occupational factors, disability prevalence, posture etc in between. I recommend collating like concepts to make a more coherent background and detailing the evidence gap for the need of this study.

Methods

Were there any exclusion criteria?

How was “higher risk of developing LBP due to long-term static sedentary work” identified? Was this identified from the selection criteria?

How were random codes devised? (line 157). The text is unclear about “gender, age” It reads as the participants were given random gender allocation and ages when the data entered into the software.

How long it did it take SpinalMouse to complete the scan?

Was the SpinalMouse device calibrated?

Were there any missing data?

How was feasibility measured?

The description of the data analysis could be strengthened and described in relation to the study’s aims.

The sample size calculation is absent.

Results

The Tables are well presented and labeled.

Subheadings would be useful to break up the block text.

The text in the 2nd paragraph is descriptive and maybe better suited reporting in table format.

Some sections of the results repeat the methods e.g. line 225-231.

Some statements are unsupported by evidence from the results e.g. “It is clearly visible that upright sitting compared to standing position increases the anterior inclination of the whole spine, decreases the lumbar curvature and anterior tilt of the pelvis, decreasing thoracic curvature. Limitation in mobility can be seen in a sitting position for U-F and E-F except the thoracic spine.”

Discussion

The third paragraph repeats the results.

Round 2

Reviewer 2 Report

Dear Authors,

Thank you for the opportunity to review manuscript entitled “Functional analysis of the spine with the Idiag SpinalMouse System among sedentary workers affected by non-specific low-back pain”.

I accept all made changes.